# Is Adherence to the Saudi Healthy Plate Dietary Guidelines Associated with Eating Concerns Symptoms among Saudi Young Females?

**DOI:** 10.3390/nu16121931

**Published:** 2024-06-18

**Authors:** Mona Mohammed Al-Bisher, Hala Hazam Al-Otaibi

**Affiliations:** Department of Food and Nutrition Science, College of Agricultural and Food Science, King Faisal University, Al-Ahsa 31982, Saudi Arabia; 218005414@student.kfu.edu.sa

**Keywords:** Saudi Healthy Plate Dietary Guidelines, food-based dietary guidelines, eating behaviors, eating concerns, Saudi young females

## Abstract

Background: Food-based dietary guidelines (FBDGs) offer broad recommendations based on scientific evidence, focusing on food groups rather than nutrients that should be included in the diet. Emerging adulthood (18–30 years) is a critical period for poor dietary quality and mental health. Eating habits (EHs) are formed early in life and are influenced by various factors, such as emotional state, which can lead to either binge or restricted eating, ultimately increasing the risk of eating disorders (EDs). This cross-sectional study aimed to investigate the extent of adherence to the Saudi Healthy Plate Dietary Guidelines (SHPDGs) and its potential association with Eating Concerns (ECs) among Saudi females (aged 18–30 years) from all provinces in the Kingdom of Saudi Arabia. Methods: A validated online questionnaire was used to assess eating behaviors (EBs) using the Starting The Conversation (STC) instrument and EC symptoms using the Eating Disorders Screen for Primary Care (ESP) screening tool. Results: The total sample size was 1092 participants with a mean age of 23.02 ± 3.47. Only 0.7% of the participants adhered to the SHPDGs and were free of EC symptoms. Conversely, 50.4% of participants who exhibited EC symptoms had poor adherence to the SHPDGs. Across Saudi Arabian provinces, high adherence to the SHPDGs was more prominent in both the Eastern and Western provinces (37.5%) than in the Central and Southern provinces (0%). The most striking result was that the Central province exhibited a high percentage of poor adherence to the SHPDGs (25.6%). Moreover, high adherence to SHPDGs was not associated with the probability of ECs. Conclusions: The present study revealed a trend of poor adherence to SHPDGs among Saudi females, with a large proportion also experiencing EC symptoms. Accordingly, the authors recommend increasing awareness within the Saudi community about SHPDGs using educational campaigns on social media platforms to enhance the importance of adopting a healthy diet, especially among females, and demonstrate that the impact on their health and well-being is that they are experiencing multiple phases that involve pregnancy and giving birth involves specific nutritional requirements.

## 1. Introduction

Special attention to nutritional status is important and constitutes a lifelong investment in overall health and well-being. A healthy diet could have a beneficial impact on health by supporting growth and reducing the risk of various chronic diseases such as cancer, heart disease, and diabetes, as well as by decreasing mortality rates [1,2]. The World Health Organization identified that a healthy diet should include a high consumption of fruits, vegetables, and whole grains while limiting the consumption of sugars, salts, processed food, and saturated fats [3]. This underscores the importance of developing food-based dietary guidelines (FBDGs) that are the cornerstone of healthy diets.

The FBDGs are a joint effort between the Food and Agriculture Organization (FAO) and the World Health Organization (WHO) to assist countries in developing their own nutritional education principles [4]. These guidelines were established based on scientific evidence of the relationship between diet and health outcomes [5]. The primary purpose of the FBDGs is to offer broad recommendations regarding the types of foods that should be included in the diet, focusing on food groups rather than nutrients [6]. Moreover, the FBDGs address public health issues [6], which can serve as a basis for developing various national policies and initiatives regarding food, nutrition, and health [7,8]. At the global level, the FBDGs differ in their respective contexts; however, they often share numerous elements, such as recommendations that advise the inclusion of a variety of foods in the diet and prioritize specific food groups over others, such as fruits, vegetables, legumes, and animal-source foods, while limiting the intake of sugar, fat, and salt [6,8,9]. The FBDGs are usually presented simply and understandably, accompanied by graphical representations [6], pyramids, and circles as the most common forms of visual approach that assists in comprehending and implementing dietary recommendations more effectively [9].

At a national level, significant attention has been paid to the health status of the Saudi Arabian population. Recently, the Saudi Food and Drug Authority (SFDA) launched the Saudi Healthy Plate Dietary Guidelines (SHPDGs), which serve as comprehensive guides offering recommendations that can be effectively utilized for food choices and the enhancement of a healthy lifestyle [10]. The SHPDGs like other international guidelines (United States (MyPlate); United Kingdom (Eatwell Guide); Australian Dietary Guidelines) emphasize a balanced diet, portion control, and the importance of fruits and vegetables. They also advise limiting intake of sugars, salts, and unhealthy fats. The SHPDGs are specifically tailored to local dietary customs, eating habits and available foods, traditional dietary practices, social norms, providing targeted advice to address prevalent health risks in Saudi Arabia, such as obesity, cardiovascular disease, and diabetes.

The Healthy Food Palm (HFP) was launched by the Ministry of Health as an FBDG for Saudis, designed in the shape of a palm tree to reflect Saudi identity [11].

Throughout every stage of life, specific nutritional requirements affect health and the risk of diseases [12]. Eating habits (EHs) are developed early in life [3,13] and have a crucial influence on food and drink preferences in adulthood, a period when the likelihood of chronic disease development is high [12]. One concern is that these habits tend to persist into adulthood, making them resistant to change until old age [13]. Recommendations derived from the FBDGs can fulfill nutritional requirements by improving food choices during that period, thereby minimizing the likelihood of developing chronic diseases [14]. Various factors influence EHs. Psychological factors such as emotional state can affect eating behaviors (EBs), leading to either binge or restricted eating [15]. Environmental factors also play an important role, including the impact of social media influencers on followers [15]. The transition from adolescence to young adulthood is characterized by rapid physiological changes, including physical growth and psychosocial development, along with the evolution of EHs [16]. Furthermore, young females experience numerous challenges in their lives, such as starting university or a job, the possibility of marrying, pregnancy, and raising children, which require a concentrated emphasis on healthy eating [17]. During that transition period, several changes in EBs occur, which are a result of food beliefs and taste preferences [18]. It appears that emerging adulthood (18–30 years) is a critical period for poor diet quality and mental health [19]. Previous studies have shown increased consumption of fast food, sugar-sweetened beverages, and soft drinks, along with a reduced intake of fruits and vegetables [20,21,22,23,24]. Physical inactivity has also been reported [25,26,27]. Unhealthy diets are among the many factors that can adversely affect the weight status of young adults [28].

It has been observed that EHs vary among generations, with apparent differences between mothers and daughters [29]. This variability can be attributed to multiple factors, and the influence of social media has emerged as a noteworthy factor driving these changes [30]. Previous research has indicated that young adults use social media platforms as their primary source of nutritional information [31,32,33]. Al-Bisher and Al-Otaibi (2022) discovered an association between the use of such platforms to seek nutritional information and an elevated likelihood of experiencing eating concerns (ECs) [34]. From this standpoint, nutritional knowledge is at the forefront. It has a profound impact as a compass that guides individuals in making prudent and health-enhancing food choices [15].

Prior Saudi research has explored the EBs of young adults and their association with several variables, such as body mass index (BMI) and sociodemographics [35,36,37]. Other Saudi studies sought to determine the extent of adherence to the HFP [23,38,39]. However, to the best of the authors’ knowledge, ECs have not been extensively examined as a serious issue within the scope of the studied variables. It is crucial to emphasize that ECs act as pivotal precursors that, if left unattended, could escalate into eating disorders (EDs) [40]. EDs are classified as serious mental illnesses characterized by severe disturbances in EBs and associated troubling thoughts and emotions, such as concerns about food, weight, or shape [41]. They are more prevalent in females, more common among younger than older individuals, and occur more frequently in females than in males [42]. Common EDs include anorexia nervosa, bulimia nervosa, and binge eating disorders [43]. These disorders can lead to malnutrition, affect various organ systems, and increase mortality rates [44].

Adherence to the FBDGs could play a protective role in lowering the likelihood of developing EDs [45,46]. Furthermore, a good-quality diet can improve mental health outcomes by decreasing depression, anxiety, and stress levels [47], thereby highlighting the critical role of diet and physical activity as predictors of mental health and well-being in young adults [48].

According to the current knowledge, there is a gap that requires further investigation. Therefore, this study aimed to achieve the following objectives:(1)Exploring the eating behaviors (EBs) of Saudi females.(2)Assessing the adherence level to the Saudi Healthy Plate Dietary Guidelines (SHPDGs).(3)Investigating the probability of an association between adherence to the Saudi Healthy Plate Dietary Guidelines (SHPDGs) and suffering from eating concerns (ECs) symptoms.

## 2. Materials and Methods

### 2.1. Study Design and Participants

A cross-sectional study was conducted in the Kingdom of Saudi Arabia between 23 November 2020 and 31 January 2021. The study population exclusively consisted of young females. A convenience sampling approach was used to select participants who met the following eligibility criteria: Saudi nationality, age 18–30 years, non-pregnant and non-lactating status, disease free, and willingness to participate in the study. Females that did not meet these criteria were excluded. The sample size was calculated using Daniel’s (1999) equation [49] with a minimum required size of 310 participants. The initial number of the sample reached 1988 participants; after sorting according to the eligibility criteria, the final number became 1092. Some participants suffered from chronic diseases; 1.7% had diabetes mellites, 1% had high blood pressure, 2.2% had genetic blood disorders (G6PD deficiency, Sickle cell anemia, Thalassemia), and 17.5% had anemia. As the rates of these participants were lower in comparison to the rates of participants who were disease-free (79%), they were included in the final sample.

### 2.2. Study Instruments

A validated questionnaire was distributed through various social media applications using Google Forms. The questionnaire comprised three sections. The first section included demographic information (e.g., age, marital status) and anthropometric measurements (e.g., weight, height) for calculating the body mass index (BMI) using the equation provided by the World Health Organization [50]. The second section explored EBs and the extent of their adherence to SHPDGs. The third section screened for symptoms of eating concerns.

#### 2.2.1. Eating Behaviors (EBs)

##### The Saudi Healthy Plate Dietary Guidelines (SHPDGs)

A national survey conducted in 2013 by the Ministry of Health in the Kingdom of Saudi Arabia reported the prevalence of obesity (28.7%) and overweight (30.7%) in Saudi society (aged ≥ 15 years) [10]. These findings are likely warning signs indicating a defect in EBs, along with a sedentary lifestyle, leading to serious health problems that must be addressed. Consequently, the relevant Saudi nutrition authorities undertook action. In a collaborative effort between the Saudi Food and Drug Authority (SFDA) and the National Nutrition Committee (NNC), SHPDGs were launched in 2020 based on extensive research and scientific studies [10]. They believed that SHPDGs could assist Saudi individuals in improving their EBs. Introduced as a visual guideline, the plate is circular and divided into five sections to facilitate public absorption. Every section of the plate illustrates the recommended amounts from each food group for a healthy and balanced daily meal, thereby aiding in the creation of healthier food choices. This dietary plate was designed according to the daily calorie requirements recommended for healthy adults, with 2000 calories for females and 2500 for males. Some key recommendations include the following [10].

Increasing the daily intake of fruits and vegetables.Staying hydrated by drinking adequate fluid per day.Engaging in exercise for at least 150 min/week is a regular healthy habit.Choosing whole grains rich in fiber instead of refined grains.Mindful fat consumption and choosing unsaturated fats in small amounts.Limiting consumption of red and processed meats.Including two servings of fish in the weekly meals.Choosing low-fat dairy products over full-fat options.Minimizing consumption of junk foods.Reading food labels while shopping and being attentive to nutritional facts to avoid products high in saturated fat, sugar, and salt.Considerations in cooking methods; grilling is considered healthier than frying.Prioritize variety; the variety within each food group is as crucial as quantity.

##### Exploration of Eating Behaviors (EBs)

Eating behaviors were explored using the Starting The Conversation (STC) tool designed by Paxton et al. (2011) [51]. It is a simple and effective instrument for dietary assessment in both the English and Spanish languages [51]. It comprises eight questions designed to screen the frequency of consumption of several food groups, including fruits, vegetables, meats, legumes, fats, and junk food, including fast food and sweet foods. Furthermore, the authors expanded the tool to include four additional questions to encompass all major food groups, including milk and dairy products, grains and bread, water intake, and physical activity habits according to the SHPDGs [10]. This comprehensive approach enabled a thorough evaluation of participants’ EBs. It is crucial to highlight that the authors (two bilingual native English-speaking translators) translated the tool into Arabic and backward translated the initial translation to make sure that the Arabic version reflected the same item contents of the original version, ensuring its alignment with Saudi eating habits and cultural norms while preserving its inherent meanings. Subsequently, they shared the translated questionnaire with multiple faculty members from the Department of Nutrition Science at the college and provided feedback to validate its content. Following this validation step, the tool’s reliability was assessed by administering it to 30 volunteers who met the eligibility criteria as the final sample. Cronbach’s alpha coefficient value of 0.7 or more for all items demonstrates a satisfactory level of internal consistency, confirming the reliability of the data collection tool. To facilitate accurate responses regarding serving sizes, the authors included visual aids such as pictures and accompanying comments in the questionnaire to help participants estimate the appropriate portion sizes. In response to each question on the STC tool, participants were assigned a score within the range of 0 to 2 (Appendix A).

##### Assessing the Adherence to the SHPDGs

According to the scale in Table 1, the total scores ranged from 0 to 24. Based on these scores, the participants were classified into three independent groups to describe their level of adherence to the SHPDGs: high adherence (0–7), moderate adherence (8–15), and low adherence (16–24).

Notably, SHPDGs have unspecified recommendations regarding junk food and physical activity. The authors considered “never” as a suggested frequency for consumption of junk food, while “daily” has been suggested for physical activity. For fats, the authors retained the options mentioned by Paxton et al. (2011) [51] in their tool and suggested “very little” for recommended consumption.

#### 2.2.2. Prevalence of Eating Concern (EC) Symptoms

This was assessed using the screening tool The Eating Disorders Screen for Primary Care ESP [52], which contains four questions presented in a “Yes” or “No” format. Every answer of “No” was considered a normal response, while “Yes” was considered an abnormal response, except for question number one (Are you satisfied with your eating patterns?), where “Yes” indicated a normal response. Obtaining two or more abnormal responses indicated suffering from eating concerns. Because there is no Arabic version of this instrument, the authors (two bilingual native English-speaking translators) converted it into Arabic using a communicative translation and backward translated the initial translation to make sure that the Arabic version reflected the same item contents of the original version. It was then revised by multiple faculty members from the Department of Nutrition Science within the college to verify its validity. Subsequently, its reliability was tested by administering it to 30 volunteers who met the eligibility criteria as the final sample, and the Receiver Operating Characteristic Curve (ROC) test was implemented. A high value (ROC = 1) was achieved, which indicates that the screening tool has high sensitivity and can identify all positive cases correctly [53].

### 2.3. Statistical Analysis

After treating outlier data and verifying that they were normally distributed, the following tests were employed: descriptive analyses, including frequencies and percentages, as well as means and standard deviations, and inferential analyses, such as chi-square for categorical variables and one-way ANOVA for continuous variables. Binary logistic regression was used to examine the association between adherence to SHPDGs and EC symptoms. All analyses were conducted using the Statistical Package for the Social Sciences (SPSS) version 23. A *p*-value ≤ 0.05 was considered statistically significant.

### 2.4. Ethical Approval

Ethical approval was obtained from the Research Ethics Committee (REC) of King Faisal University (KFU) under approval number 20988, issued on 18 November 2020.

## 3. Results

The total sample consisted of 1092 participants. Of those, 24.8% were from the Central province, followed by the Eastern, Western, Northern, and Southern provinces with 22.7%, 20.3%, 17.1%, and 15%, respectively. Furthermore, more than half the participants (56.1%) did not adhere to the SHPDGs. Only a negligible fraction of the participants (0.7%) adhered closely, whereas 43.1% adhered moderately (Table 1).

### 3.1. Sample Characteristics

Table 1 shows the sociodemographic characteristics and anthropometric measurements of the participants. The mean age was 23.02 years (SD ± 3.47) with a significant difference between the three groups at *p* = 0.017, and most participants (55.5%) had a normal BMI. More than half were unmarried (83.2%), and 87.5% were university-educated. There was a significant difference (*p* = 0.045) in adherence to the SHPDGs across the KSA, with high adherence being prominent in both Eastern and Western provinces (37.5%) compared to the Central and Southern provinces (0%). The most striking result was that the Central province exhibited a high percentage of poor adherence to the SHPDGs (25.6%).

### 3.2. Eating Behaviors (EBs) and Level of Adherence to the Saudi Healthy Plate Dietary Guidelines (SHPDGs)

Regarding the level of adherence to the SHPDGs, most participants showed a low adherence level (56.1%), whereas 0.7% of participants exhibited a high adherence level, with a mean score of 6.88; SD ± 0.35. A small minority of participants consumed fruits and vegetables in five or more portions daily (0.5% and 1.1%, respectively), and most of them adhered to the SHPDGs at a moderate level, with a significant difference among the three groups (*p* = 0.000) (Table 2).

As shown in Table 2, participants who adhered to the SHPDGs at a low level were more likely to consume fast food (10%), soft drinks (27.2%), crackers and chips (34.3%), and sweets (35.7%) three times or more weekly. Furthermore, they had more deficiencies in the daily consumption of meat (2.1%), milk and dairy products (3.3%), and water (4.2%) and were inactive (77%) compared to participants who adhered to SHPDGs at high and moderate levels.

### 3.3. Prevalence of EC Symptoms

Among the 1092 participants, 44.3% exhibited EC symptoms. According to the ESP tool, the group with low adherence to the SHPDGs recorded the highest rate of positive responses to most questions (Table 3).

### 3.4. Association between Adherence to the Saudi Healthy Plate Dietary Guidelines (SHPDGs) and Symptoms of Eating Concerns (ECs)

The binary logistic regression analysis revealed noteworthy results. Adherence to the SHPDGs at a moderate level will reduce the probability of developing EC symptoms by 42% at the significance level (*p* = 0.000) compared to adherence to a low level. According to the ESP tool criteria, the probability of dissatisfaction with eating patterns will be reduced by 69% at the significance level (*p* = 0.000), and the probability of the effect of weight on self-perception will be reduced by 39% at the significance level (*p* = 0.000) compared to adherence at a low level (Table 4).

## 4. Discussion

The main objective of the current study was to investigate the EBs of Saudi young females and assess the extent of adherence to SHPDGs and their association with EC symptoms. Most participants had unhealthy EBs and poor adherence to the SHPDGs. Nearly 56.1% of the participants exhibited low adherence to the SHPDGs, and most of them (50.4%) were at high risk of developing EDs according to the diagnostic criteria of ECs.

In this study, a small minority of participants adhered to the SHPDGs at a high level (0.7%). This result supports previous research conducted in KSA, which revealed that most Saudis did not adhere to the Saudi FBDGs mentioned in the HFP [23,38,39]. In addition, the present results corroborate those of El Ansari and Samara (2018) in Egypt and Martimianaki et al. (2022) in Greece [54,55]. This could be due to a lack of awareness regarding FBDGs. Alnasser (2023) found that the majority of participants were not aware of both FBDGs outlined in the SHPDGs and the HFP; just a small percentage of participants were familiar with those guidelines by name only, while others recognized them primarily through visual illustrations [56]. It is important to note that the recognition of names and images does not necessarily correlate with the comprehension or understanding of the guidelines themselves [56]. There are many barriers to a healthy diet among young adults, including unhealthy diets of family and friends, the widespread prevalence of unhealthy foods, sometimes at a relatively low cost, lack of knowledge and skills to prepare healthy foods, unhealthy snacking, stress eating, high prices of healthy foods, easy access to junk food, and, more importantly, a lack of motivation to eat healthily [28,57].

Interestingly, a minimal percentage (2%) of participants reported consuming five or more portions of fruits and vegetables daily (48.5% and 39.6%, respectively). In contrast, a substantial number of participants did not meet the recommended daily intake. These results are consistent with those of previous studies. For instance, in the USA, only 9% of females consume four or more portions of fruits and vegetables daily [58]. Similarly, in the KSA, just 2.78% of females achieve this level of consumption [59]. Recent research in KSA also supports the current results, with only 8% of females meeting the daily intake of five or more portions of fruits and vegetables [60]. However, these results differ from those reported in previous studies conducted by Al-Otaibi (2014) and El Ansari et al. (2015) [61,62]. Several barriers could restrict the recommended daily intake of fruits and vegetables. As highlighted by Livingstone et al. (2020), many young adults may not prefer the taste of fruits and vegetables, encounter challenges in storage and keeping them fresh, or face time constraints due to commitments such as studying or jobs [63]. In addition, some individuals may not have an adequate appetite for the recommended portions, making it less feasible to include them in their daily consumption [63]. Cost may also act as a barrier to meeting the recommended fruit and vegetable servings [64] and to healthy eating [65]. These factors may collectively explain why most participants in the current study reported consuming only 1–2 portions of fruits and vegetables per day (56.5% and 61.1%, respectively).

Protein sources, including red meat, chicken, fish, eggs, and legumes, were consumed at a low rate of three or more portions per day (5.9%) by most participants, with 59.5% reporting only one portion per day and 26% consuming two portions per day. This result supports the research by Yahia et al. (2016), who reported that 24% of females consumed just one portion per day and 9% consumed two portions per day [58]. These are potentially attributable to food preferences, which rank among the primary factors influencing food choices [66].

Only 7.6% of the participants reported consuming three or more portions per day of milk and dairy products, whereas most participants (53.9%) consumed one portion per day. It is noteworthy that Saudi studies have highlighted that adolescents and young females are at high risk for developing osteoporosis and osteopenia with aging, which is attributable to many behaviors, including low consumption of dairy products, increased consumption of soft drinks, and sedentary behavior [67,68]. Milk and dairy products provide substantial benefits to skeletal health and overall wellbeing. Bone strength is influenced by various factors including bone mass, which typically peaks at the end of the third decade of life [69]. Therefore, any nutrient deficiency, such as that of calcium (an essential mineral found in milk and dairy products), can lead to a lower peak bone mass and an increased risk of hip fractures later in life [70]. It is important to note that females undergo major physiological changes in their bodies during pregnancy and lactation, which can manifest as changes in their bone status [71].

When participants were asked about their fat consumption, the majority (55.8%) reported consuming ‘some’, which they considered as an appropriate amount based on their self-estimation. Remarkably, among those adhering to SHPDGs at low levels, more fats were consumed daily compared to those adhering to SHPDGs at high levels (3.8% and 0%, respectively). Some factors impact food choices; Yahia et al. (2016) revealed a negative correlation between nutritional knowledge and unhealthy fat and cholesterol intake, in which participants who had greater nutritional knowledge consumed low amounts of fats [72]. Notably, the current study did not include detailed questions regarding specific types of fat.

Significant differences were observed among the three groups in fast food, soft drinks, crackers, chips, and sweeteners. Given the inclination of young adults to align with contemporary trends, such findings were anticipated. They tend to prefer junk food because it gives them independence in their food choices without parental assistance. Saha et al. (2021) reported many factors that increase fast food consumption, including young adulthood, lack of nutritional knowledge, taste preferences, internet addiction, pricing, promotional offers, easy accessibility, and the reputation of fast food brands [73]. Fast food consumption is a primary risk factor for decreased diet quality and increased calorie and fat intake, and high consumption is a risk factor for overweight, abdominal obesity, diabetes, metabolic syndrome, and cardiovascular disease [74].

Consistent with previous studies, approximately 60.7% of the participants reported an absence of physical activity from their daily routines [27,59,75]. However, this finding differed from those of others [76,77]. This may be attributed to the limited availability of exercise facilities specifically designated for females in Saudi Arabia, which has been identified as a barrier [78].

One of the most remarkable findings of this study was that the group that met the SHPDG criteria (0.7%) did not exhibit any symptoms of ECs (0%). Conversely, the group with low adherence to the SHPDGs (56.1%) mostly displayed EC symptoms (50.4%).

In this study, 484 participants (44.3%) achieved ESP instrument scores of 2 or higher. This rate is significantly higher than that reported in the USA by Fruehwirth et al. (2023) (24%) but lower than that reported by Purkiewicz et al. (2021) (69%) [79,80]. The variation in the results may be related to the fact that the current study encompassed a wider age range and included more participants. The final objective of this study was to determine the association between adherence to SHPDGs and EC symptoms. High adherence to the SHPDGs was not associated with the probability of developing symptoms correlated with ECs, such as dissatisfaction with eating patterns and the impact of weight on self-perception, compared with moderate adherence to the SHPDGs. These findings highlight the pivotal association among eating behaviors, EBs, and overall well-being. Extensive research has shown that consuming adequate amounts of fruits and vegetables, ideally five or more servings daily, has a beneficial influence on health by reducing the risk of depression, elevating happiness, and increasing life satisfaction [81,82,83,84,85]. Additionally, lifestyle factors also have an impact; a poor lifestyle can affect mood [86], and mood and food choices reciprocally influence each other. Consequently, healthy food choices play a critical role in enhancing mood [87] and providing resilience against mood fluctuations [88].

This study has several strengths. To the authors’ knowledge, this study is one of the first to investigate the level of adherence to SHPDHs and its association with suffering from EC symptoms, besides the instruments (STC, ESP) that have not been applied previously in the KSA. Focusing on the age group of 18–30 years is pivotal, considering that this period includes numerous developmental milestones, particularly skeletal and fertility aspects that are influenced by dietary intake. Furthermore, the sample size was large and was taken from different provinces across Saudi Arabia, offering a comprehensive overview of the EBs of females considering cultural and environmental variations.

However, this study had some limitations. First, an association found in a cross-sectional study does not imply causation. The second limitation is the potential for bias in many regards due to the sampling procedure of convenience type, and the use of an online questionnaire, which may restrict the generalizability of the results, which were exclusively reported for females who searched for nutritional information on social media platforms. Moreover, the online questionnaire relies on self-reported data; for some data, such as height, weight, and dietary intake, it depends on the participants’ ability to recall accurately, and some may misjudge data as either overestimating or underestimating. Third, both the STC and ESP instruments were considered brief and lacked detail. Fourth, the findings should be interpreted with caution as they are specific to young Saudi females and may not apply to other populations or age groups.

## 5. Conclusions

In conclusion, this study revealed a concerning trend of poor adherence to SHPDGs among Saudi females, with a large proportion also suffering from EC symptoms. Furthermore, the present findings demonstrate a negative association between adherence to the SHPDGs and EC symptoms, such as dissatisfaction with eating patterns and the impact of weight on self-perception. These findings underscore the urgent need to prioritize efforts aimed at promoting SHPDGs and increasing awareness within the Saudi community. Strategies for achieving this may include targeted campaigns on social media platforms such as Instagram, Snapchat, and TikTok to disseminate evidence-based information, promote positive body image, and provide resources for individuals struggling with eating concerns. Additionally, integration of nutritional education into university curricula is important. It is also imperative to emphasize to young females the importance of adopting healthy EBs and their major impact on overall health and well-being, as they experience multiple phases that involve pregnancy and childbirth.

Embracing a healthy lifestyle, including a healthy diet and regular physical activity, requires strong intentions and motivation to maintain healthy behaviors. It is essential to recognize that prioritizing health requires the continuity of hard work and discipline.

To sum up, our findings provide unique insights into the association between adherence to the SHPDGs and the probability of incidence. Further research employing mixed-method approaches is needed to delve deeper into the motivations behind unhealthy EBs, ECs, and noncompliance with the SHPDGs, particularly within this demographic. Understanding these motivations can provide valuable insights for the development of more effective interventions.

## Figures and Tables

**Table 1 nutrients-16-01931-t001:** Sociodemographic characteristics and anthropometric measurements of the participants.

Variables	Total N = 1092		Group A N = 8 (0.7%)		Group B N = 471 (43.1%)		Group C N = 613 (56.1%)		
	Mean	±SD	Mean	±SD	Mean	±SD	Mean	±SD	*p*-Value
Age (Year)	23.02	±3.47	23.88	±3.72	23.34	±3.45	22.76	±3.47	0.017 *^a^
Weight (kg)	58.52	±12.90	55.13	±8.74	59.13	±12.82	58.10	±13.01	0.322 ^a^
Height (m)	1.58	±0.06	1.57	±0.05	1.58	±0.06	1.58	±0.06	0.673 ^a^
BMI (kg/m^2^)	23.38	±4.85	22.46	±3.72	23.68	±4.88	23.17	±4.84	0.202 ^a^
Marital Status	N	%	N	%	N	%	N	%	*p*-Value
Unmarried	909	83.2%	7	87.5%	389	82.6%	513	83.7%	0.846 ^b^
Married	183	16.8%	1	12.5%	82	17.4%	100	16.3%	
Level of Education	N	%	N	%	N	%	N	%	*p*-Value
High School or Below	136	12.5%	1	12.5%	50	10.6%	85	13.9%	0.275 ^b^
University or Higher	956	87.5%	7	87.5%	421	89.4%	528	86.1%	
Employment Status	N	%	N	%	N	%	N	%	*p*-Value
Unemployed	967	88.6%	7	87.5%	407	86.4%	553	90.2%	0.149 ^b^
Employed	125	11.4%	1	12.5%	64	13.6%	60	9.8%	
Province of Residence	N	%	N	%	N	%	N	%	*p*-Value
Eastern	248	22.7%	3	37.5%	117	24.8%	128	20.9%	0.045 *^b^
Central	271	24.8%	0	0%	114	24.2%	157	25.6%	
Western	222	20.3%	3	37.5%	108	22.9%	111	18.1%	
Northern	187	17.1%	2	25%	74	15.7%	111	18.1%	
Southern	164	15%	0	0%	58	12.3%	106	17.3%	
Perceived Health Status	N	%	N	%	N	%	N	%	*p*-Value
Good	657	60.2%	7	87.5%	319	67.7%	331	54%	0.000 **^b^
Average	395	36.2%	1	12.5%	139	29.5%	255	41.6%	
Poor	40	3.7%	0	0%	13	2.8%	27	4.4%	
Dieting	N	%	N	%	N	%	N	%	*p*-Value
No	916	83.9%	5	62.5%	362	76.9%	549	89.6%	0.000 **^b^
Yes	176	16.1%	3	37.5%	109	23.1%	64	10.4%	
BMI Categories	N	%	N	%	N	%	N	%	*p*-Value
Underweight (BMI < 18.5)	155	14.2%	1	12.5%	62	13.2%	92	15%	0.732 ^b^
Normal (18.5 ≤ BMI ≤ 24.9)	606	55.5%	5	62.5%	258	54.8%	343	56%	
Overweight (25 ≤ BMI ≤ 29.9)	214	19.6%	2	25%	93	19.7%	119	19.4%	
Obesity (BMI ≥ 30)	117	10.7%	0	0%	58	12.3%	59	9.6%	

Group A (high adherence to the SHPDGs); Group B (moderate adherence to the SHPDGs); Group C (low adherence to the SHPDGs); ^a^ ANOVA test; ^b^ chi-square test; * *p* ≤ 0.05; ** *p* ≤ 0.001. Abbreviations: N, sample size; %, percentage; SD, standard deviation; kg, kilogram; m, meter; BMI, body mass index.

**Table 2 nutrients-16-01931-t002:** Eating behaviors (EBs) and level of adherence to the Saudi Healthy Plate Dietary Guidelines (SHPDGs) among the participants.

Variables	Total N = 1092		Group A N = 8 (0.7%)		Group B N = 471 (43.1%)		Group C N = 613 (56.1%)		
Daily Consumption of Fruits	N	%	N	%	N	%	N	%	*p*-Value
≥5 Servings	5	0.5%	0	0%	5	1.1%	0	0%	0.000 **^b^
3–4 Servings	60	5.5%	5	62.5%	40	8.5%	15	2.4%	
1–2 Servings	617	56.5%	2	25%	314	66.7%	301	49.1%	
Never	410	37.5%	1	12.5%	112	23.8%	297	48.5%	
Daily Consumption of Vegetables	N	%	N	%	N	%	N	%	*p*-Value
≥5 Servings	12	1.1%	0	0%	10	2.1%	2	0.3%	0.000 **^b^
3–4 Servings	81	7.4%	4	50%	57	12.1%	20	3.3%	
1–2 Servings	667	61.1%	3	37.5%	316	67.1%	348	56.8%	
Never	332	30.4%	1	12.5%	88	18.7%	243	39.6%	
Daily Consumption of Protein Sources	N	%	N	%	N	%	N	%	*p*-Value
≥3 Servings	64	5.9%	5	62.5%	46	9.8%	13	2.1%	0.000 **^b^
2 Servings	284	26%	3	37.5%	161	34.2%	120	19.6%	
1 Serving	650	59.5%	0	0%	229	48.6%	421	68.7%	
Never	94	8.6%	0	0%	35	7.4%	59	9.6%	
Daily Consumption of Grains and Bread	N	%	N	%	N	%	N	%	*p*-Value
3–4 Servings	422	38.6%	6	75%	245	52%	171	27.9%	0.000 **^b^
1–2 Servings	632	57.9%	2	25%	210	44.6%	420	68.5%	
Never	38	3.5%	0	0%	16	3.4%	22	3.6%	
Daily Consumption of Milk and Dairy Products	N	%	N	%	N	%	N	%	*p*-Value
≥3 Servings	83	7.6%	2	25%	61	13%	20	3.3%	0.000 **^b^
2 Servings	247	22.6%	3	37.5%	136	28.9%	108	17.6%	
1 Serving	589	53.9%	2	25%	209	44.4%	378	61.7%	
Never	173	15.8%	1	12.5%	65	13.8%	107	17.5%	
Daily Consumption of Fats and Oils	N	%	N	%	N	%	N	%	*p*-Value
Very little	455	41.7%	7	87.5%	277	58.8%	171	27.9%	0.000 **^b^
Some	609	55.8%	1	12.5%	189	40.1%	419	68.4%	
A lot	28	2.6%	0	0%	5	1.1%	23	3.8%	
Daily Consumption of Water	N	%	N	%	N	%	N	%	*p*-Value
≥6 Cups ^c^	164	15%	5	62.5%	133	28.2%	26	4.2%	0.000 **^b^
5–6 Cups ^c^	252	23.1%	3	37.5%	149	31.6%	100	16.3%	
2–4 Cups ^c^	451	41.3%	0	0%	134	28.5%	317	51.7%	
<2 Cups ^c^	225	20.6%	0	0%	55	11.7%	170	27.7%	
Weekly Consumption of Fast Food	N	%	N	%	N	%	N	%	*p*-Value
Never	267	24.5%	7	87.5%	197	41.8%	63	10.3%	
1–3 Times	760	69.6%	1	12.5%	270	57.3%	489	79.8%	0.000 **^b^
≥4 Times	65	6%	0	0%	4	0.8%	61	10%	
Weekly Consumption of Soft Drinks	N	%	N	%	N	%	N	%	*p*-Value
Never	476	43.6%	8	100%	329	69.9%	139	22.7%	
1–2 Times	431	39.5%	0	0%	124	26.3%	307	50.1%	0.000 **^b^
≥3 Times	185	16.9%	0	0%	18	3.8%	167	27.2%	
Weekly Consumption of Crackers and Chips	N	%	N	%	N	%	N	%	*p*-Value
Never	83	7.6%	3	37.5%	67	14.2%	13	2.1%	
1 Time	338	31%	4	50%	213	45.2%	121	19.7%	0.000 **^b^
2–3 Times	397	36.4%	1	12.5%	127	27%	269	43.9%	
≥4 Times	274	25.1%	0	0%	64	13.6%	210	34.3%	
Weekly Consumption of Sweet Food	N	%	N	%	N	%	N	%	*p*-Value
Never	140	12.8%	5	62.5%	104	22.1%	31	5.1%	
1–3 Times	670	61.4%	3	37.5%	304	64.5%	363	59.2%	0.000 **^b^
≥4 Times	282	25.8%	0	0%	63	13.4%	219	35.7%	
Physical Activity	N	%	N	%	N	%	N	%	*p*-Value
Daily	90	8.2%	4	50%	72	15.3%	14	2.3%	
3–4 Times weekly	339	31%	4	50%	208	44.2%	127	20.7%	0.000 **^b^
Never	663	60.7%	0	0%	191	40.6%	472	77%	
Eating Behavior Scores	Mean	±SD	Mean	±SD	Mean	±SD	Mean	±SD	*p*-Value
	15.55	±2.85	6.88	±0.35	13.06	±1.83	17.58	±1.41	0.000 **^a^

Group A (high adherence to the SHPDGs); Group B (moderate adherence to the SHPDGs); Group C (low adherence to the SHPDGs); ^a^ ANOVA test; ^b^ chi-square test; ^c^ one cup equivalent to 200 mL; ** *p* ≤ 0.001. Abbreviations: N, sample size; *%*, percentage; SD, standard deviation.

**Table 3 nutrients-16-01931-t003:** Prevalence of eating concern (EC) symptoms among the participants.

Variables	Total N = 1092		Group A N = 8 (0.7%)		Group B N = 471 (43.1%)		Group C N = 613 (56.1%)		
ESP Question 1	“Are you satisfied with your eating patterns?”								
	N	%	N	%	N	%	N	%	*p*-Value
Yes	319	29.2%	8	100%	198	42%	113	18.4%	0.000 **^b^
No	773	70.8%	0	0%	273	58%	500	81.6%	
ESP Question 2	“Do you ever eat in secret?”								
	N	%	N	%	N	%	N	%	*p*-Value
No	917	84%	8	100%	400	84.9%	509	83%	0.325 ^b^
Yes	175	16%	0	0%	71	15.1%	104	17%	
ESP Question 3	“Does your weight affect the way you feel about yourself?”								
	N	%	N	%	N	%	N	%	*p*-Value
No	637	58.3%	7	87.5%	305	64.8%	325	53%	0.000 **^b^
Yes	455	41.7%	1	12.5%	166	35.2%	288	47%	
ESP Question 4	“Do you currently suffer with or have you ever suffered in the past with an eating disorder?”								
	N	%	N	%	N	%	N	%	*p*-Value
No	952	87.2%	8	100%	407	86.4%	537	87.6%	0.467 ^b^
Yes	140	12.8%	0	0%	64	13.6%	76	12.4%	
ESP Classification	N	%	N	%	N	%	N	%	*p*-Value
ESP < 2	608	55.7%	8	100%	296	62.8%	304	49.6%	0.000 **^b^
ESP ≥ 2	484	44.3%	0	0%	175	37.2%	309	50.4%	
Total ESP Scores	Mean	±SD	Mean	±SD	Mean	±SD	Mean	±SD	*p*-Value
	1.41	±1.04	0.13	±0.35	1.22	±1.08	1.58	±0.98	0.000 **^a^

Group A (high adherence to the SHPDGs); Group B (moderate adherence to the SHPDGs); Group C (low adherence to the SHPDGs); ^a^ ANOVA test; ^b^ chi-square test; ** *p* ≤ 0.001. Abbreviations: N, sample size; %, percentage; SD, standard deviation; ESP, Eating Disorders Screen for Primary Care; every answer with “No” has a score of 0, while every answer with “Yes” has a score of 0, except the first question, where “Yes” has a score of 0 and “No” has a score of 1.

**Table 4 nutrients-16-01931-t004:** Association between adherence to the Saudi Healthy Plate Dietary Guidelines (SHPDGs) and symptoms of eating concerns (ECs).

Variables ^d^		B	S.E.	Wald	df	Sig.	Exp(B)
ESP Classification	Low ^e^			18.801	2	0.000	
	High	−21.219	14,210.361	0.000	1	0.999	0.000
	Moderate	−0.542	0.125	18.801	1	0.000 **	0.582
ESP Question 1	Low ^e^			69.498	2	0.000	
	High	−22.690	14,210.361	0.000	1	0.999	0.000
	Moderate	−1.166	0.140	69.498	1	0.000 **	0.312
ESP Question 2	Low ^e^			0.703	2	0.704	
	High	−19.615	14,210.361	0.000	1	0.999	0.000
	Moderate	−0.141	0.168	0.703	1	0.402	0.869
ESP Question 3	Low ^e^			17.289	2	0.000	
	High	−1.825	1.072	2.898	1	0.089	0.161
	Moderate	−0.487	0.126	14.990	1	0.000 **	0.614
ESP Question 4	Low ^e^			0.335	2	0.846	
	High	−19.248	14,210.361	0.000	1	0.999	0.000
	Moderate	0.105	0.182	0.335	1	0.563	1.111

^d^ Binary logistic regression; ^e^ reference category; ** *p* ≤ 0.001. Abbreviations: B, unstandardized regression coefficient; S.E., standard error; df, degrees of freedom; Sig., significant; Exp(B), exponentiation of the B coefficient.

## Data Availability

The datasets used and analyzed during the current study are available from the corresponding authors upon reasonable request due to privacy.

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
