# Peer review of "Is Adherence to the Saudi Healthy Plate Dietary Guidelines Associated with Eating Concerns Symptoms among Saudi Young Females?"

_nutrients, 2024, doi:10.3390/nu16121931_

Round 1

Reviewer 1 Report

Comments and Suggestions for Authors

Dear Authors,

The manuscript (nutrients-3047539) submitted for review is quite interesting and I recommend the manuscript for revision.

Authors, Please note and address the following comments:

The topic of the manuscript submitted for review is interesting.

Presented keywords (Saudi Healthy Plate Dietary Guidelines; Food-Based Dietary Guidelines; Saudi Dietary Guidelines; Eating Behaviors; Eating Concerns; Eating Disorders; Saudi Young Females) should be corrected, because they do not reflect the content of the manuscript well.

The introduction is well written.

Lines 186-188: The authors wrote” „The resulting Cronbach's alpha coefficient demonstrated a satisfactory level of internal consistency, confirming the reliability of the data collection too”. What was the value of the Cronbach coefficient?

The study is well designed. The number of respondents selected for the study is large enough and the research tools are appropriate.

Research results are presented and explained appropriately.

The sections such as material and methods, results, discussion, as well as conclusion are very well written.

References: References are not cited in accordance to Nutrients journal rules.

Despite my comments, I am pleased to recommend the manuscript for revision. I believe that it concerns an important area of research in an international context.

 Reviewer

Author Response

Thank you for taking the time to review our manuscript and for your insightful comments. We appreciate the expertise and effort you have invested in evaluating our work. Below (attached file), we address your comments in detail.

Reviewer 2 Report

Comments and Suggestions for Authors

Abstract:

·      The study's findings are limited to young Saudi females, which may not be generalizable to other populations or age groups. Revise conclusion statement accordingly.

·      The cross-sectional nature of the study restricts the ability to establish causal relationships between adherence to the SHPDGs and ECs.

·      Keep abstract concise. Too much of data is not necessary in this section, highlight the key findings.

Introduction:

·      Some information is repeated, such as the importance of FBDGs, which could be consolidated for clarity.

·      While the introduction discusses the importance of healthy diets, it lacks specific details on the SHPDGs compared to other international guidelines.

·      More emphasis on the unique cultural and societal factors in Saudi Arabia that may influence eating behaviors and adherence to dietary guidelines would strengthen the context.

Materials and Methods:

·      The use of convenience sampling may limit the representativeness of the sample, potentially introducing selection bias.

·      While the translation process of the tools is described, more details on the validation process of the Arabic versions would be beneficial.

·      The use of the STC tool and the ESP screening tool, while validated, may not capture the full complexity of EBs and ECs, potentially limiting the study's depth.

Results & discussion:

·      There is insufficient discussion of potential confounding variables that could affect the relationship between adherence to SHPDGs and ECs.

·      The discussion should clearly differentiate between correlation and causation, given the cross-sectional study design.

·      While comparisons with previous studies are made, there is a lack of depth in discussing why these differences or similarities might exist.

·      The recommendations for increasing awareness and educational campaigns are broad. More specific, actionable strategies would strengthen the practical implications of the study.

·      The manuscript would benefit from tighter editing to remove redundancy and enhance clarity.

·      Incorporating a theoretical framework on how dietary guidelines and ECs are linked would provide a stronger foundation for the study.

·      Could the data be presented in a more visually intuitive way, such as using histograms or other graphical representations, rather than just placing the data in tables?

Author Response

(The authors gave the same response as above.)

Reviewer 3 Report

Comments and Suggestions for Authors

This is an interesting research study with adequate novelty. However, some points should be addressed.

- The authors should add subheadings in the Abstract based on the guidelines of the journal (e.g. Background, Methods, Results, Conclusions).

- The abbreviation KSA did not explained in the Abstract. Please specify.

- In line 39, instead "diabetes" the authors could better report "metabolic disorders".

- At the last sentence of the 1st paragraph of the Introduction, the authors should add a relevant reference.

- In the 4th paragraph, in the 2nd sentence, the authors should more relevant and validated references beyond reference 12 (lines 68-70).

- The introduction is very well written and well organized, emphasizing the literature gap that the present study aims to cover.

- In section 2.1, Please specify if you excluded or included females wich suffered from any diseases.

In section 2.1, a statement with the final response rated should be reported. In fact the authors should report the initial number of participants and the final number of participans.

- Because there are a lof of Tables into the manuscript, the Table 1 could be included as supplementary material. From my point of view, the inclusion of Table 1 into the main text is not wrong but it may reduce the readability of the paper.

- In section 2.2, the authors report that "A validated questionnaire was distributed through various social media applications using Google Forms". This statement should also be included in the Abstract. If the completion of the questionnaire was performed online this should be emphasized and it should also be included in the limitations of the present study.

- In section 2.3, the authors should include more details for the statistical analysis of their data. Do they use a normality test for the continious variables. A statement concerning the potential confouder factors should also be added.

- In section 2-2-1-1, the authors should add relevant reference, e.g. lines 146, 151, 157.

- In section 2-2-1-2, the authors should add a relevant reference, e.g. line 180

- The section 3 is a bit complex for the readers and the authors shoul improve the syntax of this section.

- If the study was perform by online questionnaire this should be reoirted in the limitation of the study, as the online filling questionnaires are not so reliable compared to those filling by face-to-face interviews.

Comments on the Quality of English Language

Moderate editing of English language required

Author Response

(The authors gave the same response as above.)

Round 2

Reviewer 2 Report

Comments and Suggestions for Authors

I appreciate authors for addressing all the comments with appropriate answers.

I also appreciate authors for making substantial changes where necessary, this version of manuscript deserves publication.